# Radiation Damage on Silicon Photomultipliers from Ionizing and Non-Ionizing Radiation of Low-Earth Orbit Operations

**DOI:** 10.3390/s24154990

**Published:** 2024-08-01

**Authors:** Stefano Merzi, Fabio Acerbi, Corinne Aicardi, Daniela Fiore, Vincent Goiffon, Alberto Giacomo Gola, Olivier Marcelot, Alex Materne, Olivier Saint-Pe

**Affiliations:** 1Fondazione Bruno Kessler (FBK), Center for Sensors and Devices, Via Sommarive 18, 38123 Trento, Italy; acerbi@fbk.eu (F.A.); gola@fbk.eu (A.G.G.); 2Centre National d’Études Spatiales (CNES), 18, Avenue Edouard Belin, 31076 Toulouse, France; corinne.aicardi@cnes.fr (C.A.); alex.materne@cnes.fr (A.M.); 3AIRBUS Defence and Space, 31, Rue des Cosmonautes, 31076 Toulouse, France; daniela.fiore@airbus.com (D.F.); olivier.saintpe@airbus.com (O.S.-P.); 4Institut Supérieur de l’Aéronautique et de l’Espace (ISAE-SUPAERO), 10, Avenue Édouard Belin, 31076 Toulouse, France; vincent.goiffon@isae-supaero.fr (V.G.); olivier.marcelot@isae-supaero.fr (O.M.)

**Keywords:** silicon photomultiplier, SiPM, single-photon, SPAD, proton, X-ray, radiation damage

## Abstract

Silicon Photomultipliers (SiPMs) are single photon detectors that gained increasing interest in many applications as an alternative to photomultiplier tubes. In the field of space experiments, where volume, weight and power consumption are a major constraint, their advantages like compactness, ruggedness, and their potential to achieve high quantum efficiency from UV to NIR makes them ideal candidates for spaceborne, low photon flux detectors. During space missions however, SiPMs are usually exposed to high levels of radiation, both ionizing and non-ionizing, which can deteriorate the performance of these detectors over time. The goal of this work is to compare process and layout variation of SiPMs in terms of their radiation damage effects to identify the features that helps reduce the deterioration of the performance and develop the next generation of more radiation-tolerant detectors. To do this, we used protons and X-rays to irradiate several Near Ultraviolet High-Density (NUV-HD) SiPMs with small areas (single microcell, 0.2 × 0.2 mm^2^ and 1 × 1 mm^2^) produced at Fondazione Bruno Kessler (FBK), Italy. We performed online current-voltage measurements right after each irradiation step, and a complete functional characterization before and after irradiation. We observed that the main contribution to performance degradation in space applications comes from proton damage in the form of an increase in primary dark count rate (DCR) proportional to the proton fluence and a reduction in activation energy. In this context, small active area devices show a lower DCR before and after irradiation, and we propose light or charge-focusing mechanisms as future developments for high-sensitivity radiation-tolerant detectors.

## 1. Introduction

Silicon photomultipliers (SiPMs) are solid state photon detectors composed of arrays of passively-quenched Single-Photon Avalanche Diodes (SPADs), also referred to as a “microcell”, all connected in parallel and working above breakdown condition in Geiger mode [1]. These detectors have gained increasing interest in the past decade in many fields such as medical imaging [2], automotive LiDAR (Light Detection And Ranging) [3], Cherenkov telescopes [4], and big physics experiments [5]. In recent years, the continuous improvement of SiPM performance, combined with their advantages like ruggedness, compactness, lightweight and low operating bias, as well as high photon detection efficiency from the ultraviolet to the near infrared, opened new possibilities for these detectors in space applications, such as satellite operations for cosmic ray studies [6,7], atmospheric LiDAR [8], and navigation LiDAR.

When SiPM are used in these kinds of applications, as well as in others like high-energy physics experiments [9,10,11], one major problem is that they are exposed to high radiation doses that degrade their performance over time due to damage caused by both ionizing and non-ionizing radiation effects [12] (especially increasing their primary noise, i.e., the dark count rate). This causes an increase in power consumption and a reduction in photon detection efficiency (PDE) due to the high cell occupancy, which can severely impact the experiment during its lifetime. Ionizing radiation generally causes damages by inducing interface states at the silicon surface, while non-ionizing radiation induces defects in the bulk of the silicon detector by displacement damage. Several studies have been conducted in recent years to evaluate the effects of radiation damage on SiPM functionality [12,13,14,15,16,17,18,19,20,21]. Considering the types of particles and the fluences and doses that are representative of most space missions [13,17], the major contributor to SiPM deterioration is due to non-ionizing energy loss (NIEL). Most of the literature reports that the main effect of radiation damage is an increase in dark count rate (DCR) proportional to the fluence while minor consensus was found on other parameters, such as signal amplitude, gain, and photon detection efficiency (up to fluences of about 10^11^ n_eq_/cm^2^).

At Fondazione Bruno Kessler (FBK), Italy, we conducted a series of experiments with protons and X-rays, as reported in [18,19,20,21], using several SiPM produced over the years with different versions of layout and based on different process technologies. While these works provided insightful information on the behaviour of SiPMs and the worsening of their performance after irradiation, it was not straightforward to isolate the effect of a single layout and process variation on the effects of radiation damage on their performance.

In this paper, we present a systematic study of radiation damage on SiPM, of both NIEL and IEL. We tested several process and layout splits of FBK SiPMs, based on NUV-HD technology produced specifically for irradiation studies. The SiPMs were irradiated with protons and X-rays and characterized before, during and after irradiations. The aim is to easily compare and isolate the effect of a single feature variation on the SiPM response to radiation and highlight possible development paths for future radiation-tolerant SiPM technologies.

For the proton irradiation level, we chose fluences covering most of the Earth observation and scientific space missions (LEO, GEO, and Lagrange points). The total radiation dose to the detectors varies depending on the type of orbit, on the mission lifetime and on the shielding of the devices in the satellites. A reasonable estimate was calculated for LEO in [17,20] with estimated fluences in the order of 10^11^ n_eq_/cm^2^, which was used as a reference for this study as a worst-case scenario among most usual EO and scientific orbits, due to the impact of radiation belts on the radiation environment.

On the other hand, regarding ionizing radiation, for low earth orbits a total dose in the order of 10^2^ Gy/year is expected for a 1 mm Al shield [22]. More generally, for optical detectors integrated into space optical instruments and sensors operated in LEO, GEO and Lagrange points, the ionizing dose at mission end of life is of the order of a few 100 Gy(Si). However, previous works showed little to no deterioration of SiPM performance at these doses and, in order to investigate the effect of ionizing radiation on SiPMs, we irradiated the devices up to 10^5^ Gy.

## 2. Devices under Test

The devices studied in this work are part of a dedicated SiPM lot, produced in FBK and based on NUV-HD technology [23] with several layout versions and process variations to investigate different aspects of the devices and how they correlate with the radiation-induced performance worsening. The NUV-HD technology was chosen, as its PDE spectrum peaks in the blue and NUV wavelength region, matching the most common inorganic scintillator emission spectra (used in many medical imaging machines, high-energy physics experiments and calorimeters for space-experiments), as well as NUV LiDAR LASERs wavelengths, thus making it the most promising candidate in space operation.

Regarding the process splits, we focused on studying the effects of the doping profiles in the SPADs and the BEOL (back-end-of-line) dielectrics, such as:Different electric field profiles and peak values inside the SPADs, to mitigate field enhancement effects on DCR increase after irradiation [24],Different internal doping profiles in the SPADs,Different anti reflective coatings (ARC), particularly the ones used for Near Ultraviolet (NUV) and Vacuum Ultraviolet (VUV) sensitive SiPMs, as reported in [23,24,25].

The SiPMs are produced on 3 × 4 mm^2^ silicon dies, each containing 12 detector structures divided into three detector sizes (single microcell, 0.2 × 0.2 mm^2^ and 1 × 1 mm^2^) with four microcell sizes each (15, 20, 30 and 40 µm). We focused mainly on 1 × 1 mm^2^ devices, which allow us to have a measurable DCR level before and after irradiation, and to have a reasonable statistic on noise and detection efficiency of the SPADs that compose the SiPM. In addition to the 1 × 1 mm^2^, we included smaller size SiPMs as well as a single microcell to perform a finer waveform analysis after irradiation, when the DCR is too high to precisely discriminate individual photo peaks, even on the 1 × 1 mm^2^ SiPMs. For all these SiPMs, we implemented layout splits focused on the following:The distance between the active area (high field region) and the deep trenches to study their possible contributions to the DCR increase.Rectangular-shaped microcells (as a variation to the common square shape) to study possible contributions of border effects.A microcell with a metal-ring masking the dead region outside the active area of each microcell [26], to investigate possible field plate effects on the charges generated during the irradiation.

In addition to these devices, we also tested some SiPMs produced with NUV metal-in-trench (NUV-MT) technology, which differentiate from the other ones since the deep trench isolation between microcells are filled with metals instead of just dielectrics, as detailed in [27], with a size of 1 × 1 mm^2^ and single SPADs, with two microcell sizes of 16 µm and 40 µm with metal masking.

To identify the devices, we named them using the wafer number in the production lot, which also defines the process split, alongside additional characters to define the process split. The naming convention is described in Table 1 and Table 2.

We irradiated some selected chips from each of these wafers, and we characterized them before and after irradiation, with the measurement of primary noise (i.e., dark count rate, DCR), gain, optical crosstalk probability (CT), afterpulse probability (AP), and photon detection efficiency (PDE). See [1,28,29] for details on the typical characterization procedures. Moreover, during the irradiation campaigns, we performed online measurements of the reverse current-voltage curves (I-V) of the 1 × 1 mm^2^ devices after each irradiation step, both in the dark and under constant illumination conditions.

## 3. Experimental Setup

Tested devices have been mounted on a custom Printed Circuit Board (PCB), designed to read-out up to 96 channels from 8 die sites, with 12 bonded devices per die. As can be seen in Figure 1, the dies in one of the PCB are placed on a half-circle, whose diameter has been chosen to be fully included in the estimated uniform-fluence region as well as the uniform X-ray dose region at the two irradiation facilities, respectively.

Two of these PCBs are used simultaneously during the irradiation (i.e., mounted together on the support and exposed simultaneously during irradiation), with one placed below the other and rotated at 180 degrees. In this way, it is possible to use the full uniform fluence region and double the number of irradiated devices. The first PCB undergoes all irradiation steps, and the IVs of 32 channels (1 × 1 mm^2^ SiPMs) are measured after each irradiation step. Conversely, in the second slot, the PCBs are not connected to the readout for online measurements, but they will be tested offline after the irradiations. During the irradiation campaign, several copies of these PCBs are packaged and irradiated at different dose levels. In this way, we were able to have both online measurements of IV curves from the same samples at all fluences (doses), as in the irradiation described in [21], as well as the possibility to have dies irradiated at several fluencies (doses) for offline full characterization through waveform analysis, as in the irradiation campaign described in [18].

The complete setup for irradiation is composed of four main parts, shown in Figure 2 on the left:Carrier board for the PCBs to be irradiated.Black box to ensure darkness during IV measurement.Motorized stage, open during irradiation to avoid partial stopping of the beam, closed to ensure darkness during IV measurement.Blue LED for IV measurement under illumination.Measurements system, composed by a National-Instrument Switching matrix with 32 channels in a 4 × 8 configuration (four channels measured simultaneously) and source-and-measurements unit.Power supplies for controlling the LED and the motorized stages.Local computer for the control of the instrumentation and remote computer for user control of the setup during the irradiation.

Figure 2 on the right shows the inside of the black box on the carrier PCB with the position of the two PCBs to be irradiated. The primary PCB, measured online, is at the bottom, and it is connected to the headers for the 96 channels, while the secondary PCBs are placed at the top and can be easily replaced between irradiation steps.

## 4. Irradiation Facilities and Steps

### 4.1. Proton Irradiation

The proton irradiation campaign was held at the proton therapy facility in Trento [30] in April 2023. The irradiation took place in the research line using the “dual ring” setup, with uniform beam intensity over a circle with a diameter of 60 mm. The proton beam is emitted at an energy of 148 MeV, which was lowered down to 74 MeV using 10 panels of RW3 [31] (1 cm thick). The proton beam current can be tuned between 1 nA and 300 nA. The total amount of charge, proportional to the number of protons generated during each irradiation step, is measured through a Lynx ionization chamber [32]. This was also used before irradiation to verify the shape and uniformity of the beam. The proton flux in this configuration was measured at 3.84 × 10^5^ p × s^−1^ × cm^−2^ × nA^−1^.

Irradiation of the devices was carried out in 12 steps for two days. Each step had an exponentially increasing fluence compared to the previous one to span a wide range of fluences with exponentially spaced sampling. The primary PCB, placed on the bottom and measured online, was called PCB-A. Secondary PCBs are named PCB-B-X and PCB-C-X, where X is a progressive number of each copy of the PCB. Each secondary PCB is irradiated for only two steps (of the PCB-A) and then replaced with a different one. Figure 3 on the left shows the total fluence as a function of the irradiation steps for PCB-A and for the secondary PCBs. As can be seen, in the semi-logarithmic scale, the points are evenly spaced in fluence with a factor of 2.3 between each step.

Unfortunately, due to a malfunctioning of the measurement setup during the two highest fluence irradiation steps, the IV curves just after irradiation were not measured correctly. It was possible to extrapolate the data of the last point (highest fluence) by interpolating annealing data of current vs. time with an exponential and projecting back the current at t = 0. However, due to possible inaccuracy in such a procedure, these data have been used for the visual indication on the plots, but not for more quantitative analysis.

### 4.2. X-ray Irradiation

The X-ray irradiation campaign was held at the TIFPA (Trento Institute for Fundamental Physics and Applications) in Trento between 22 May 2023 and 24 May 2023. The irradiation used an X-ray tube with a tungsten anode, biased at 40 kV, with a 180 µm aluminium filter in front. It produced a 33° uniform dose region with main lines between 7.6 keV and 12 keV, and Compton emission up to 40 keV. The dose rate was adjusted by changing the current between 10 mA and 25 mA, and it was measured using an ionization chamber (PTW 30010 Farmer). This instrument was previously calibrated (conversion factor) to have the precise dose in silicon (the one reported in the paper). The tested SiPM were placed 20 cm from the tube. The dose rate in this configuration was measured to be 12.5 Gy × s^−1^ × mA^−1^.

Irradiation of the devices was carried out in 17 steps over three days. As for the proton irradiation, each step has an exponentially increasing fluence (excluding the last steps that followed a linear trend) compared to the previous one to span a wide range of fluences while maintaining an exponentially spaced sampling. As for the proton irradiation, the bottom PCB, called PCB-A, was fixed and measured online, whereas PCB-B-X and PCB-C-X (top) were irradiated only for a few steps and then replaced with a different one. Figure 3 on the right shows the total dose in silicon as a function of the irradiation steps for the main PCB and for the secondary ones.

## 5. Results of Proton Irradiation

### 5.1. Breakdown Voltage and Reverse Current

From the online measurements, specifically IV curves measured under illumination, we first extrapolated the breakdown voltage (V_BD_) using the second logarithmic derivative method (as used and reported in [18]) and we compared the current in-dark condition both at 5 V below breakdown (called “non-multiplied current”) and 3 V above breakdown (i.e., 3 V of “excess bias”), respectively. The above breakdown is called “multiplied current”.

The top of Figure 4 shows that the breakdown voltage does not change with fluence (up to the maximum level reported) for all layout and process splits. Devices with a breakdown voltage around 36 V are ULF technologies, while LF technologies have a breakdown voltage around 31 V. Also, the below breakdown, shown in the middle of Figure 4, does not show significant variation. Small variations are visible at the highest fluences.

The bottom of Figure shows the current measured 3 V above the breakdown (multiplied current). For fluences up to 4 × 10^7^ n_eq_/cm^2^, there is mostly no or only a slight change in multiplied current. Some devices show a random increase in current with many of them being annealed between irradiation steps. For fluences above 1 × 10^9^ n_eq_/cm^2^, a linear increase of current was observed with a dose, with no major difference in behaviour between devices. Between these two fluences, there is a “transition region” with a mostly linear increase of current with fluence and with strong variation between samples and irradiation steps. The main trend observed between samples is that devices with a smaller fill factor (FF, i.e., the ratio between the active area and the total area of the microcell) have about a ten times smaller current than devices with a larger FF due to the reduction in the high field volume, possibly further reduced by border effects, which in turns reduces the DCR, which is likely mainly due to field-enhanced Shockley-Read-Hall (SRH) generation [33].

### 5.2. Dark Count Rate

Different cell sizes have different gain values, so to remove this contribution from the comparison, the dark count rate for the different SiPMs is calculated as a function of the fluence. DCR is calculated from the multiplied current knowing the gain (G) and the Excess Charge Factor (ECF) according to Equation (1):(1)$$DCR=\frac{I}{q*G*ECF}$$
where q is the elementary charge. Under the assumption that the gain and the ECF do not change with irradiation (which was verified in [21]), it is possible to calculate the product of gain and ECF, also known as Gain Current (GC) from current and DCR measurements of non-irradiated devices, and use this value to estimate the DCR after irradiation, when direct measurement through waveform analysis is no longer possible due to the high noise of the detectors.

Figure 5 shows the calculated DCR as a function of the proton fluence for the different structures. It mostly follows the same behaviour of the multiplied current, with structures with a very low active area (AA 30 µm-D8) showing a DCR about ten times lower compared to other structures. This is related to the lower fill factor of these structures, and consequently, to the size of the high field region compared to the cell size: for the same defect density, there is a lower total number of defects in the high field region of low AA structures compared to other structures with the same cell size, resulting in a lower value of DCR. For the same reason, a direct relationship between cell size and DCR is observed, where smaller cells sizes, due to the smaller FF, also show a smaller DCR (Figure 5, bottom).

### 5.3. Damage Factor

To consider the presence of gain and excess charge factor in a SiPM, the damage factor α is calculated for each fluence point according to Equation (2):(2)$$\alpha =\frac{∆DCR*q}{\Phi *V*P_{t}}=\frac{∆I}{\Phi *V*P_{t}*\left(G*ECF\right)}$$
where Φ is the proton fluence, V is the volume of the high field region and P_t_ is the triggering probability of the avalanche reported in the work by [34]. This term is necessary to consider that not all damage generated by a proton result in an avalanche, and this avoids underestimating the damage parameter, especially at low bias when the triggering probability is low.

The top of Figure 6 shows the calculated damage factor for all devices as a function of the fluence at 3 V of excess bias. For fluences below 8 × 10^8^ p/cm^2^, there is a high uncertainty in the calculation due to the “random” behaviour of DCR at low fluences explained in the previous section. Above this value, the damage factor is constant with fluence, excluding the highest point in which there might be either saturation of the detectors or errors in the reconstruction of its pre-annealing currents. All the devices show a damage factor of around 10^−15^ A/cm and the differences between devices are small. This is due to the high-field region volume parameter (V), calculated from the FF of the microcell and the thickness of high-field region (neglecting field effects at the border of the active area or in the low-field depleted region). The lack of major differences in the damage factor between different devices suggests that the damage factor, in first approximation, is not easily modified with process or layout splits, and improvements in the performance of SiPM after irradiation might need to focus on reduction of the high-field region in combination with light focusing (e.g., microlens) or charge focusing mechanisms to increase the PDE of the detector while reducing the radiation-sensitive area.

The bottom of Figure 6 shows the damage factor calculated at different excess bias for the highest fluence with direct measurements (not reconstructed). It shows higher values (~10^−15^ A/cm at 3 V of excess bias) compared to the one expected for detectors without gain (3.5 × 10^−17^ A/cm) and an increase in damage factor with the excess bias. This indicates the presence of field-enhancement effects on the generation rate of the carriers. Extrapolating the damage parameters at lower biases, so at a lower value of electric field, these values approach the ones reported in literature, ranging from 2 × 10^−16^ and 9 × 10^−16^ A/cm at 0 V of excess bias.

### 5.4. Photon Detection Efficiency

In Figure 7 at the top, it is reported that the measured and estimated PDE at 3 V of excess bias for a wavelength of 420 nm for the different samples. In general, the PDE is proportional to the FF with 40 µm cells showing the highest PDE. VUV-LF devices show a lower PDE compared to NUV-LF ones due to their different ARC optimized for VUV sensitivity that negatively affects NUV and visible sensitivity. On the other hand, on NUV-ULF devices, a slightly higher PDE compared to NUV-LF is observed, likely related to the different electric field of the junction, which causes a faster increase in PDE at a low excess bias. Cells with increased distance between trench and active area have lower FF and lower PDE. For the 30 µm cells, the PDE of S0 is 40%, which reduces to 4% for the AA D8 die.

The PDE measured before irradiation is used to calculate the behaviour of PDE after irradiation by using the detector response measured through the IV curves taken under illumination at different fluences. There are some fluctuations in PDE as a function of the fluence but there seems to be no significant change in PDE, even at high fluences.

### 5.5. Gain, Crosstalk and Afterpulse

In Figure 7, we report the gain, crosstalk probability, and afterpulse probability measured at 4 V of excess bias before and after irradiation. Gain and AP were measured on SPADs at room temperature, and CT measurement was performed on 1 × 1 mm^2^ at room temperature and on 0.2 × 0.2 mm^2^ SiPMs at −20 °C after irradiation in order to discriminate single events in high DCR conditions. Since these measurements require waveform analysis, it was not possible to acquire data at each irradiation step, so the only data available are before irradiation and after irradiation at the highest dose. At the same time, it was not possible to correctly measure small cells at low excess bias due to the high electronic noise of the setup. For this reason, it is chosen to show the behaviour of Gain, CT and AP at 4 V of excess bias instead of 3 V, as in the results calculated from the IV measurements.

No significant change in gain is observed after irradiation, apart from some fluctuation caused by the uncertainty of the measurement on samples with high noise level. For CT measurements, time constraints and instability of the amplifiers operated at low temperatures limited the number of tested devices, but the results do not show a link between radiation damage and a change in crosstalk probability.

In the case of AP measurements, a significant increase for almost all devices after irradiation is observed. This effect is related to the defects generated in the silicon by displacement damage dose. The defects act as trapping centres for charges, thus increasing the AP probability for irradiated samples. A few samples do not show this increase, but this is mainly attributed to limitations in the measurement setup and software, which is not able to detect AP events below a certain amplitude and will underestimate their number in case of cells with a long recharge time.

Despite the increase in AP, its contribution to the total ECF is limited, as its probability remains relatively low. Moreover, the gain and the CT do not show a significant variation in samples measured before and after irradiation. This proves the validity of the hypothesis used in the calculation of the DCR, where it was assumed that the gain current, i.e., the gain multiplied by the excess charge factor, does not change with proton irradiation.

### 5.6. Activation Energy

To calculate the activation energy (Ea) of the SiPMs, the IV measurements at different temperatures were taken. At the same time, the temperature dependence of the breakdown voltage on temperature was also measured. For irradiated samples, the IVs were taken at temperatures between 243 K and 293 K with 2 K steps to avoid annealing of the samples. For non-irradiated samples, on the other hand, the IV measurements were taken at temperatures between 293 K and 313 K, to have higher currents and allow a better analysis of the data. Because of the high variability in Ea data, due to uncertainty in the current measurements for devices with low gain, we averaged this measurement over the four structures of each die to improve the readability of the results.

Figure 8 shows the calculated activation energy for the 32 structures before and after irradiation. Before irradiation, the samples showed an activation energy between 0.6 and 0.9 eV with no clear correlation between process splits. For irradiated samples, LF structures have a lower Ea, around 0.37 eV, while ULF structures show a slightly higher activation energy, at around 0.4 eV. This is reflected in the temperature difference needed to halve the dark current (and DCR). Before irradiation, the halving temperature is around 7 K, while after irradiation, it is ~12.3 K for LF structures; it is reduced to ~11.5 K for ULF structures. Looking into the processes for dark current generation, activation energies between 0.6 eV and 0.9 eV can be attributed to the diffusion dark current coming from interface states, while lower activation energies between 0.4 eV and 0.7 eV could be attributed to the generation of dark current from bulk defects [35].

## 6. Results of X-ray Irradiation

### 6.1. Breakdown Voltage and Reverse Current

The top of Figure 9 shows the breakdown voltage as a function of the total dose in silicon for the different structures. As in the case of proton irradiation, the breakdown voltage does not change with fluence for all layout and process splits. Small variations in breakdown, observed for all devices, are likely related to small temperature changes of the irradiation room, with a lower temperature in the morning that increases during the day.

The middle of Figure 9 shows the leakage current measured 5 V below the breakdown. No change is observed up to about 1000 Gy. For higher doses, a more than linear increase in current was observed for all devices with the faster increase measured for W6-S0 and W16-S0 devices (NUV-LF and NUV-ULF).

The bottom of Figure 9 shows the dark current, measured 3 V above the breakdown. As in the previous case, no change is observed up to 1000 Gy. For higher doses, a fast increase in dark current was observed for all NUV (W6, W8, W16) devices excluding the one with the metal masking (M*): W16 ULF shows the highest current that reaches saturation of the detector while W6-AA shows lower currents due to the small active area of the detector. On the other hand, VUV devices (W2, W4) and devices with metal masking (W6-M*) show a much slower increase in dark current with the dose.

### 6.2. Dark Count Rate

Dark count rate was calculated using the same method described above. Figure 10 shows the DCR as a function of the total dose for the different structures. All devices show a DCR around or below 10^5^ c.p.s./mm^2^ at room temperature before irradiation. Lower DCR is observed for devices with small AA, due to the smaller FF.

After irradiation, devices from the VUV process split and NUV samples with metal masking show a relatively low DCR with less than an order of magnitude increase, below 10^6^ c.p.s./mm^2^. The opposite behaviour was observed for NUV-ULF samples, with saturation of the detectors above 5 × 10^4^ Gy, and DCR above 10^9^ c.p.s./mm^2^. NUV samples without metal masking show a higher DCR that decreases with the reduction of the active area and the increase of the distance between the trench and the high field region. While partially hidden by the sample-to-sample variability, an inverse relation between cell size and DCR is present, with smaller cells showing higher DCR after irradiation (Figure 10, bottom). This is the opposite of what is observed in the case of proton irradiation, and it is likely due to charges accumulated in the insulating oxide in the trenches: smaller cells have longer trench perimeter per unit area and this can result in a higher DCR.

### 6.3. Photon Detection Efficiency

Figure 11 at the top shows the estimated PDE at 420 nm as a function of the X-ray dose. A few outliers were not plotted in Figure 11, mostly from the NUV-ULF and NUV-LF samples, which show a strong reduction in PDE above 10^4^ Gy likely due to the high DCR that causes saturation of the detectors at high doses. For all other samples, a small reduction in PDE was observed with the dose, in the order of 10% relative to the initial value, for all structures independent of the technology or layout split. A possible explanation can be the accumulation of fixed charge in the oxide layer at the surface of the detector, which can alter the electric field of the active region and reduce the PDE. Charge accumulation in the trenches, on the other hand, does not seem to affect the PDE, as there is no difference in behaviour between samples with different distances between the trench and high field region. Further investigation needs to be done to exclude possible setup issues, such as instability or degradation of the LED used for the measurement.

### 6.4. Gain, Crosstalk and Afterpulse

Figure 11 reports the gain, crosstalk probability and afterpulse probability measured at 4 V of excess bias before and after irradiation. As in the case of proton irradiation, the measurement method limits both the number of doses available and the excess bias at which the measurement is taken. The limited time availability before irradiation did not allow us to completely characterize the samples before irradiation, and their parameters were estimated from measurements performed on similar devices (represented with dashed lines in the plots) while the parameters after irradiation were measured directly. In general, no significant change in the parameters was observed with the dose, but an increased variability in their value was observed after irradiation. This is mainly attributed to the high electronic noise of the setup, which worsens the single photoelectron detection capability and, consequently, the calculation of these three parameters. One exception is the reduction of gain observed for all layout splits of W06-AA, with 30 µm cells with a reduced active area. It is possible that the accumulated charge in the trench oxide changed the shape of the electric field and the depletion region, reducing the gain of the cells, but more studies are needed to confirm this effect. A second exception is given by the increased AP probability of W02-M0*, devices with metal masking outside the active area which show an absolute increase in AP between 2% and 5% for all layout splits. These devices are from the VUV-LF process split with the standard AP surface configuration, and it is possible that the presence of the metal masking for in VUV structures might increase the sensitivity of the detector to surface TID damage, which results in an increased AP probability. More studies need to be performed to confirm the results and better understand the effect of the metal masking on the device surface.

As in the case of protons, these measurements prove the validity of the initial hypothesis used in the calculation of the DCR, where it was assumed that the Gain current does not change with irradiation.

## 7. Conclusions

We irradiated and compared the results of the performance on 32 SiPMs, with layout and process variations. We compared SiPM produced with VUV-LF, NUV-LF and NUV-ULF technology (having different ARC and electric field profiles), and with different layouts of the microcell composing the SiPM—in particular the S0, D2, AA and TS layouts, characterized by an increased distance between trench and active area and increased perimeter of the high field region.

In the first experiment, we irradiated the SiPMs with 74 MeV protons up to 1 × 10^11^ p/cm^2^, or 1.5 × 10^11^ 1 MeV n_eq_/cm^2^. No relevant change in breakdown voltage, non-multiplied current or PDE was observed up to the highest fluence. We measured a linear increase in multiplied current and DCR starting at ~10^8^ p/cm^2^ for all devices, without a major difference in damage factor between different process and layout splits. The most noticeable difference in behaviour after irradiation was observed for AA structures, where the small active area results in a lower value of DCR after irradiation due to the smaller microcell fill factor. This confirms the strong contribution of the high electric field on primary noise generation in the irradiated SPADs. Moreover, we can infer that the reduction of the high field region coupled with light or charge-focusing mechanisms seems to be a promising way to mitigate the radiation damage effects and improve the radiation tolerance of the SiPMs while maintaining a high PDE.

We also noticed that the irradiated devices showed a lower activation energy compared to before irradiation. In this case, ULF, thanks to its lower peak electric field, showed a higher activation energy compared to LF structures, resulting in a faster reduction of DCR with cooling. This effect, together with the higher PDE, measured a lower excess bias, suggesting that using ULF structures might have an advantage compared to LF ones in the production of radiation tolerant detectors. However, since this technology is still experimental and not fully developed, extensive tests are needed to properly characterize ULF devices and verify the stability of the production process.

In the second experiment, we irradiated 32 SiPMs with X-rays from a Tungsten target at 40 kV up to 10^5^ Gy. The SiPMs were produced with VUV-LF, NUV-LF and NUV-ULF technologies, with standard and low AP (having different ARC, electric field, and surface doping configuration) and with S0, AA and M0* microcell layouts (characterized by an increased distance between the trench and active area and a masking of the area outside the high field region with a metal layer). No change in breakdown voltage was observed up to the highest dose. A non-linear increase in both non-multiplied and multiplied current with dose was measured for all structures, especially for doses above 1000 Gy, with the largest increase observed for NUV-LF and NUV-ULF structures. VUV-LF structures and NUV-LF-M0* structures showed a smaller increase. First analysis suggests that smaller cells show a larger increase in DCR, probably related to the higher trench perimeter per unit area. Compared to proton irradiation, multiplied current and DCR does not increase as much at most Earth observation and scientific space mission orbit X-ray dose levels (few 10 s to 100 s of Gy), thus showing that TID is not the main contributor to DCR when compared to displacement damage dose. A small reduction in PDE as a function of the dose is observed for all devices, possibly due to charge accumulation in the oxide layer.

## Figures and Tables

**Figure 1 sensors-24-04990-f001:**
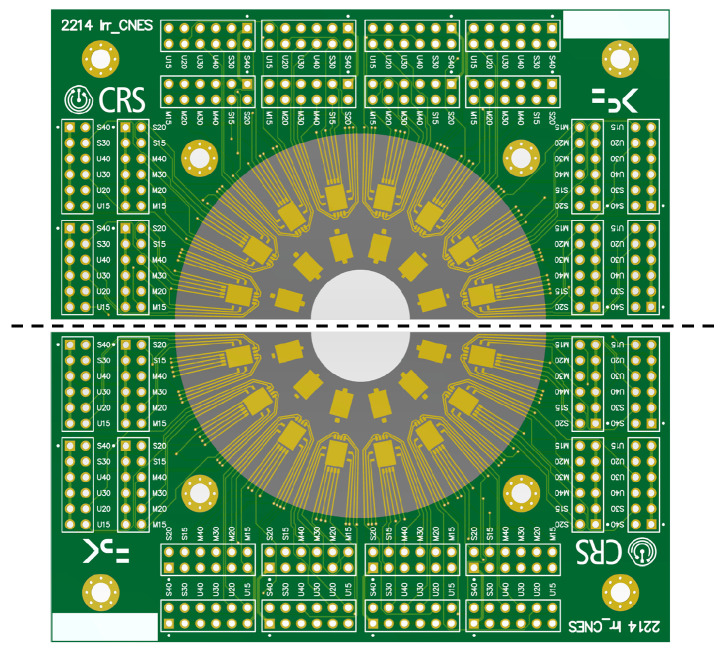
PCBs used for irradiation tests. The full setup is composed of two PCBs assembled to cover the full irradiation field.

**Figure 2 sensors-24-04990-f002:**
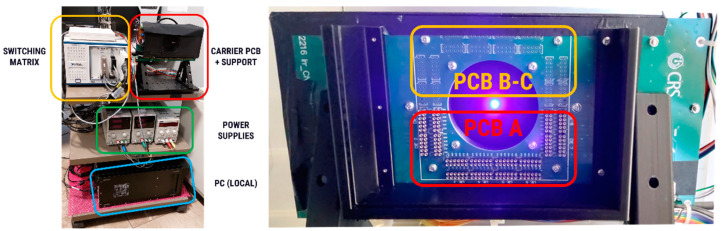
(**Left**): complete irradiation setup with the four main parts highlighted. (**Right**): detail of the carrier PCB with motorized shutter and blue LED, highlighting the positioning of the two PCBs with the samples to be irradiated. Note: the system is seen here from the backside. The LED is mounted on the shutter and, when closed, illuminates the SiPMs from the front; i.e., the same direction of the incoming proton or X-ray beam.

**Figure 3 sensors-24-04990-f003:**
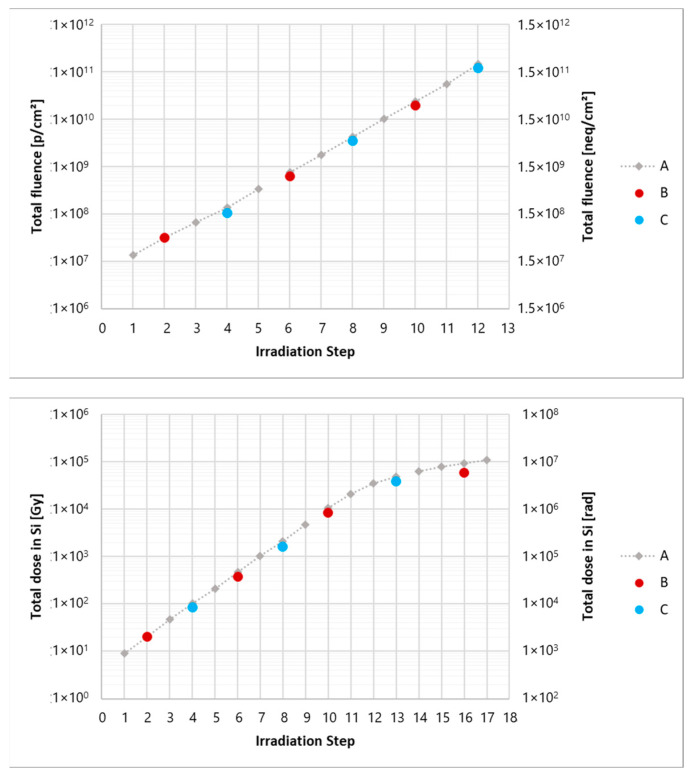
(**Top**): Total fluence as a function of the irradiation steps for the different PCBs under proton irradiation. (**Bottom**): Total dose in silicon as a function of the irradiation steps for the different PCBs under X-ray irradiation. In both cases, A is the primary PCB, measured online, while B and C are two versions of the secondary PCBs, replaced between irradiation steps.

**Figure 4 sensors-24-04990-f004:**
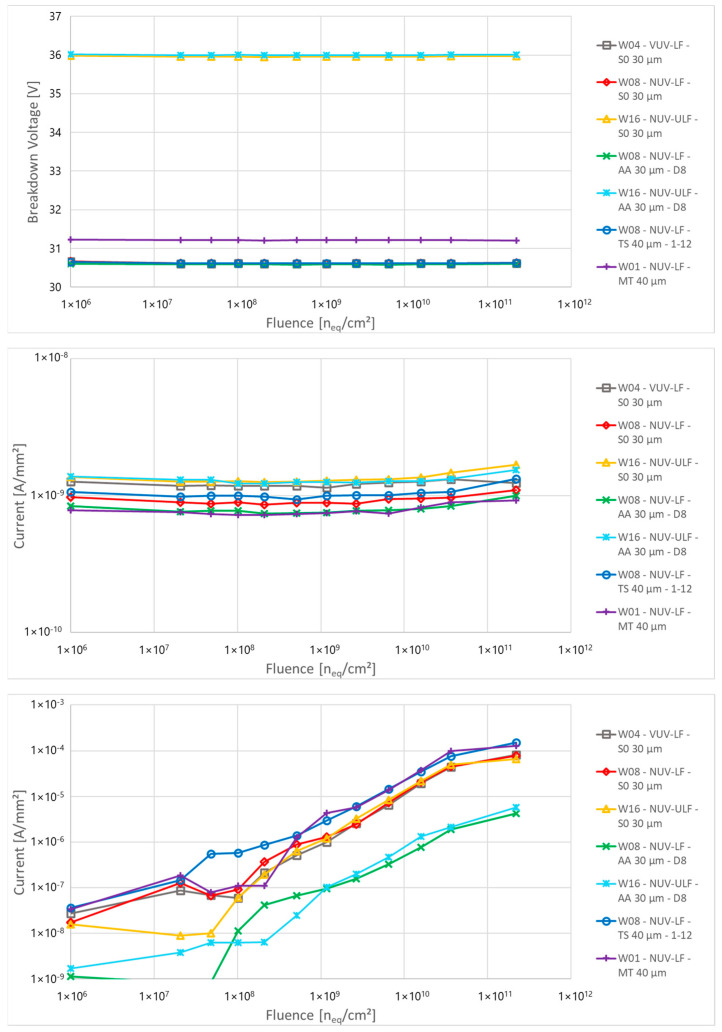
(**Top**) to (**bottom**): breakdown voltage, non-multiplied current (5 V below breakdown) and multiplied current (3 V above breakdown) as a function of the proton fluence for different structures of PCB-A.

**Figure 5 sensors-24-04990-f005:**
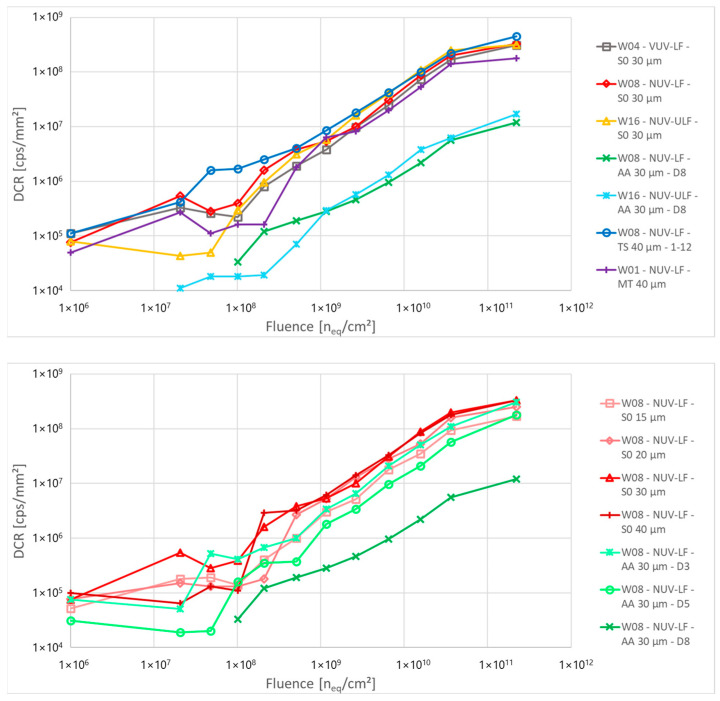
Calculated DCR as a function of the proton fluence for different structures of PCB-A, calculated at 3 V of excess bias. The bottom plot highlights the difference in DCR between cells with different sizes and different FF.

**Figure 6 sensors-24-04990-f006:**
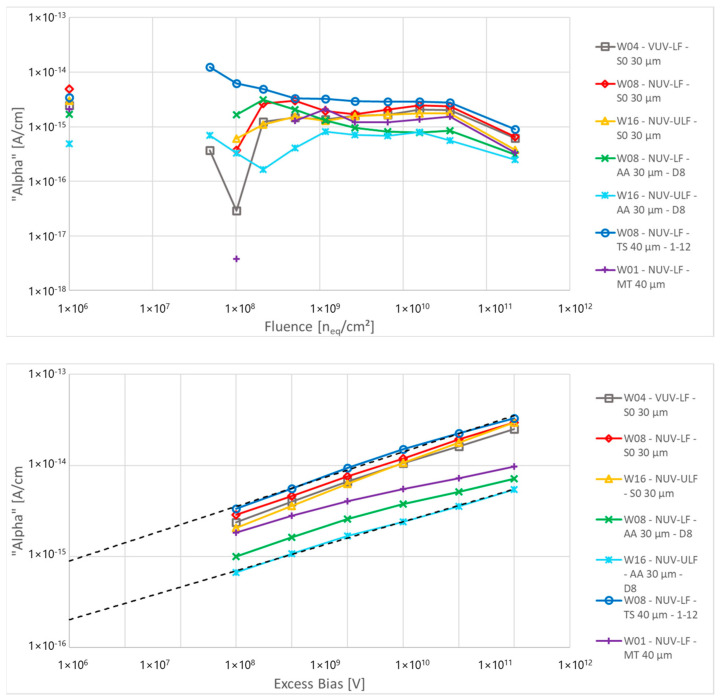
Damage parameter as a function of the proton fluence (**Top**) at 3 V of excess bias and as a function of the excess bias at 2.4 × 10^10^ p/cm^2^ (**Bottom**).

**Figure 7 sensors-24-04990-f007:**
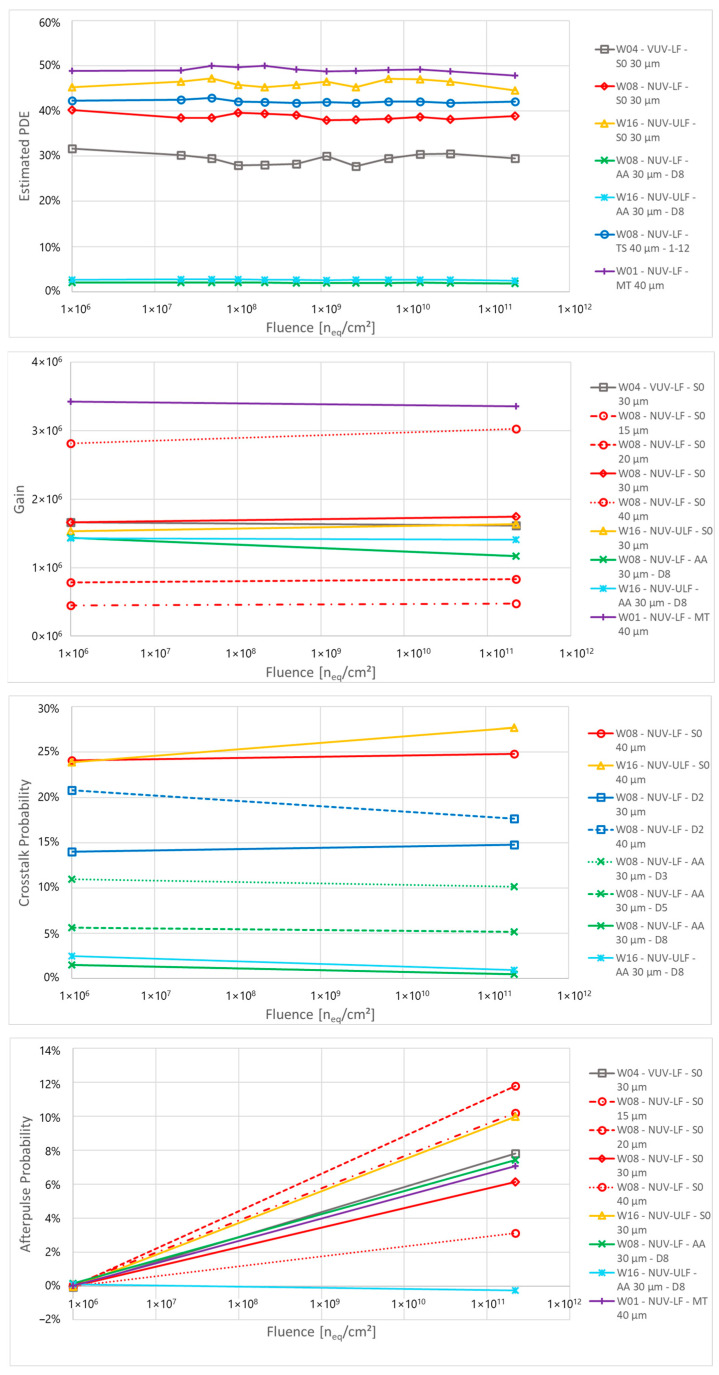
(**Top**) to (**bottom**): PDE as a function of the proton fluence for the 32 structures of PCB-A, calculated at 3 V of excess bias, gain, crosstalk probability and afterpulse probability measured before and after proton irradiation at 4 V of excess bias.

**Figure 8 sensors-24-04990-f008:**
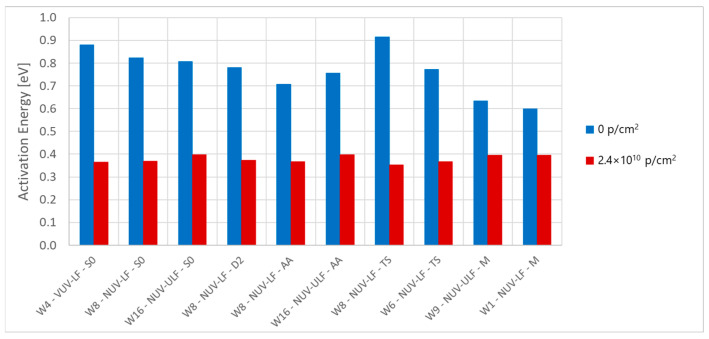
Average activation energy for the different dies calculated at 3 V of excess bias before (blue) and after (red) proton irradiation.

**Figure 9 sensors-24-04990-f009:**
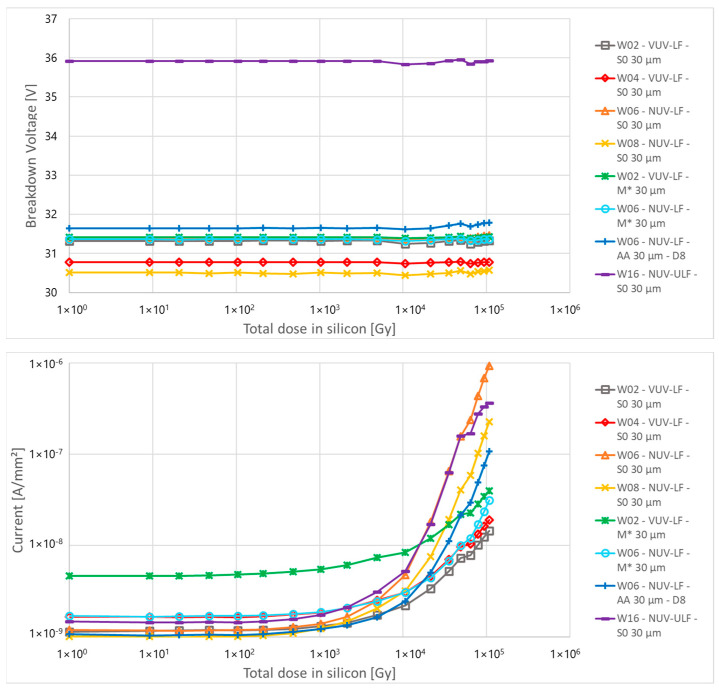
(**Top**) to (**bottom**): breakdown voltage, non-multiplied current (5 V below breakdown) and multiplied current (3 V above breakdown) as a function of the total dose in silicon for different structures of PCB-A irradiated with X-rays.

**Figure 10 sensors-24-04990-f010:**
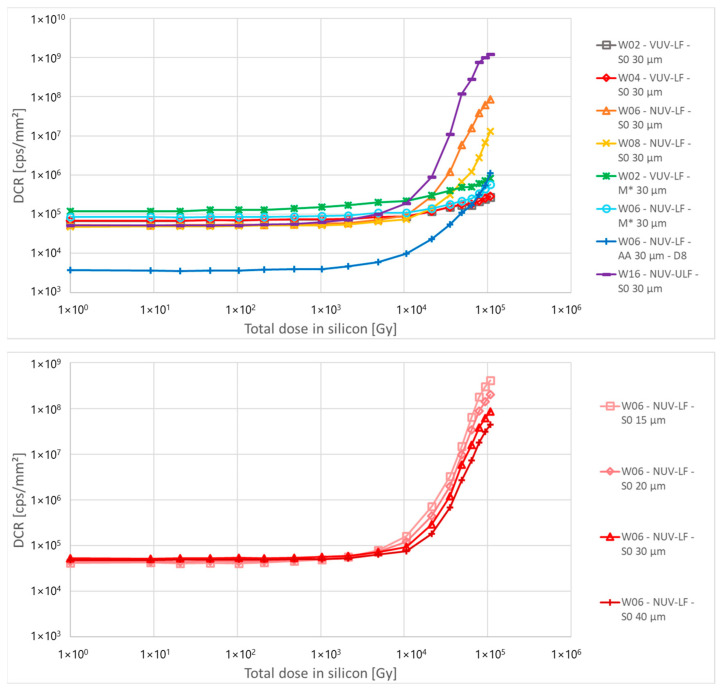
Calculated DCR as a function of the X-ray dose for different structures of PCB-A, calculated at 3 V of excess bias. The bottom plot highlights the difference in DCR between cells with different sizes.

**Figure 11 sensors-24-04990-f011:**
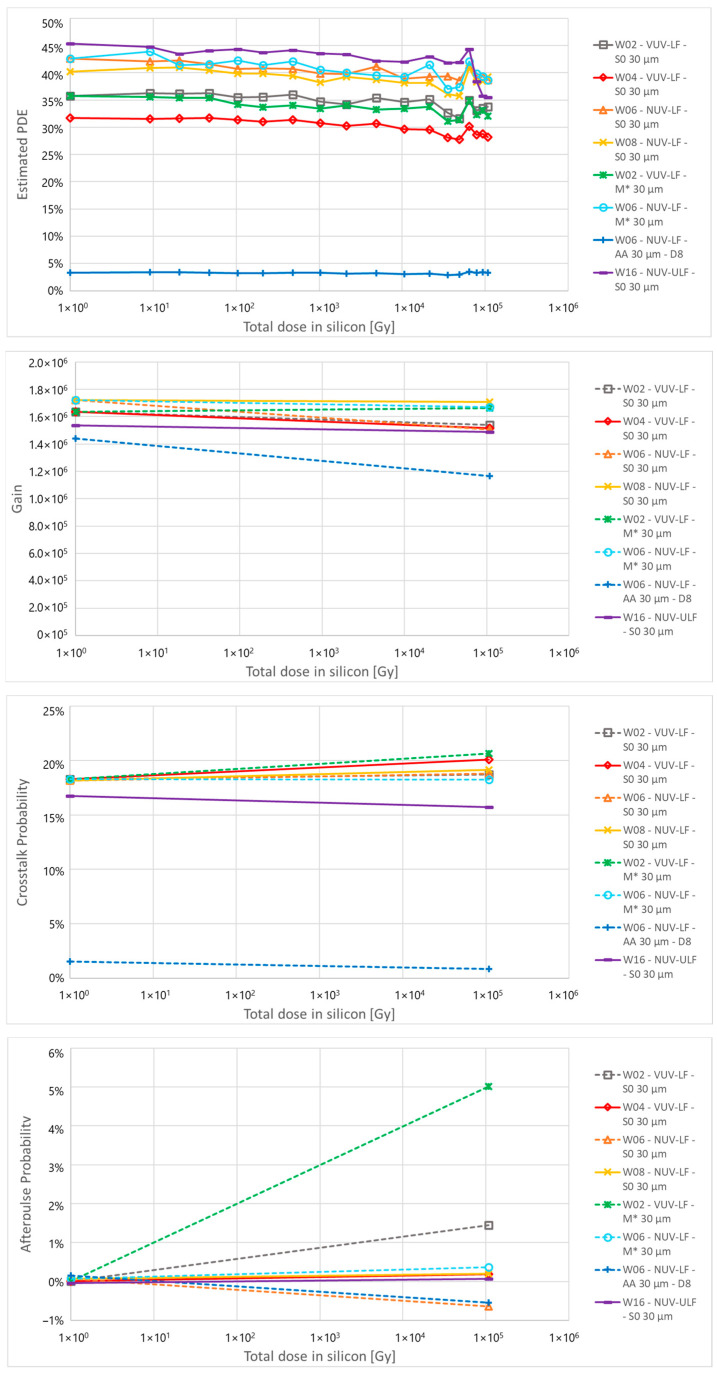
(**Top**) to (**bottom**): PDE as a function of the X-ray dose for different structures of PCB-A calculated at 3 V of excess bias, gain, crosstalk probability, and afterpulse probability measured before and after X-ray irradiation at 4 V of excess bias.

**Table 1 sensors-24-04990-t001:** Naming and features of the process splits tested in this work.

Process Split	Electric Field	Afterpulse	ARC
W02	Low (LF)	Standard	VUV
W04	Low (LF)	Low	VUV
W06	Low (LF)	Standard	NUV
W08	Low (LF)	Low	NUV
W16	Ultra-low (ULF)	Low	NUV
W01 (MT)	Low (LF)	Low	NUV
W09 (MT)	Ultra-low (ULF)	Low	NUV

**Table 2 sensors-24-04990-t002:** Naming and features of the layout splits tested in this work.

Layout Split	Notes
S0	Standard layout and microcell features
D2	Two-times increased distance between active area and trench
AA-DX	X-times (3, 5, 8) increased distance between active area and trench
M*	Metal mask outside the active area
TS	Rectangular cell: 30 µm 1–6 aspect ratio, 40 µm 1–12 aspect ratio

## Data Availability

The original contributions presented in the study are included in the article, further inquiries can be directed to the corresponding author.

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
