# Peer review of "Radiation Damage on Silicon Photomultipliers from Ionizing and Non-Ionizing Radiation of Low-Earth Orbit Operations"

_sensors, 2024, doi:10.3390/s24154990_

Round 1

Reviewer 1 Report

Comments and Suggestions for Authors

The paper addresses a relevant and timely topic with potential significant impact on space experiments. However, it requires revisions for clarity, completeness, and technical rigor. I recommend major revisions before the paper can be accepted for publication.

1. The abstract provides a concise summary of the work, highlighting the importance of SiPMs in space experiments, the challenges posed by radiation, and the objective of the study to identify design features that enhance radiation tolerance. However, it could be improved for clarity and readability by addressing the following points: Mention the types of radiation (protons and X-rays) earlier for better context. Clearly state the outcome or findings in the abstract.

2. The introduction should clearly define the problem statement, significance of the research, and the objectives. It should provide a comprehensive background on SiPMs, their applications in space, and the challenges posed by radiation exposure.

3. The methodology section should describe the experimental setup, irradiation procedures, and measurement techniques in detail. The choice of NUV-HD SiPMs and their specific configurations should be justified. Details on the irradiation doses and conditions are necessary for reproducibility.

4. The results section should present data clearly with appropriate figures and tables. Comparisons between different process and layout variations should be well-documented, highlighting the key findings. Statistical analysis of the data would strengthen the conclusions.

5. The conclusion should summarize the key findings and their implications for the development of more radiation-tolerant SiPMs. It should also suggest future research directions.

6. English language needs improvement.

Comments on the Quality of English Language

Author Response

Dear Reviewer,

Thank you very much for your work and for the opportunity to improve and refine our manuscript.
We addressed and accepted all the comment. Below we outline our response to each comment.

Comment 1: The abstract provides a concise summary of the work, highlighting the importance of SiPMs in space experiments, the challenges posed by radiation, and the objective of the study to identify design features that enhance radiation tolerance. However, it could be improved for clarity and readability by addressing the following points: Mention the types of radiation (protons and X-rays) earlier for better context. Clearly state the outcome or findings in the abstract.

Response 1: The abstract was partially rewritten by adding the main outcomes of the paper and future perspectives of this research. We do not think mentioning protons and X-ray earlier in the abstract could help, since the focus of the paper is on SiPM and their performance.

Comment 2: The introduction should clearly define the problem statement, significance of the research, and the objectives. It should provide a comprehensive background on SiPMs, their applications in space, and the challenges posed by radiation exposure.

Response 2: We modified parts of the introduction to better explain the problems caused by radiation damage, the significance of the studies related to the applications and the goal of the research. In the first two paragraphs it can be found a discussion with reference on SiPM background and their application, also in the field of space operations, and the literature on radiation damage on SiPMs. 

Comment 3: The methodology section should describe the experimental setup, irradiation procedures, and measurement techniques in detail. The choice of NUV-HD SiPMs and their specific configurations should be justified. Details on the irradiation doses and conditions are necessary for reproducibility.

Response 3: A detailed description of the experimental setup can be found at the end of section 3. We added more details explaining the choice of NUV-HD and the variations introduced on the process and on the layout, with discussion on expected effect on radiation damage response.

As for the measurement technique, we mentioned them in the second paragraph of section 3, with references to previous FBK papers and irradiation campaigns. The cited papers describe in detail the measurement techniques and the data analysis. As their discussion is quite long it was decided to not include it in this paper and to focus on the results.

A detailed discussion of the irradiation doses and conditions can be found in section 4.1 and 4.2, where we reported all information available to us.

Comment 4: The results section should present data clearly with appropriate figures and tables. Comparisons between different process and layout variations should be well-documented, highlighting the key findings. Statistical analysis of the data would strengthen the conclusions.

Response 4: In the results sections we decided to show only part of the results (6 or 7 curves out of 32) to improve readability and compare the differences between variations. We included the most noticeable devices, with the most interesting variations to better highlight the differences in response to radiation damage.
Discussion of the results and comparisons, with the main findings and suggestions for future research, are given across the result section. We prefer to keep the discussion in line with the results to reduce the complexity of the reading, as many results are extracted from the measurements and it would be confusing to group all the discussion at the end.

Since there is only a single sample for each variation and irradiation dose, it is not possible to perform statistical analysis on the results, as most of them are independent from sample to sample.

Comment 5: The conclusion should summarize the key findings and their implications for the development of more radiation-tolerant SiPMs. It should also suggest future research directions.

Response 5: We revised and improved the conclusion section to be more focused on the results achieved and highlight the suggestions for future research on the topic.

Comment 6: English language needs improvement.

Response 6: We improved the English of the manuscript as suggested.
Moreover, we think that, as this paper is submitted as an open access in a special issue, it will undergo a final English revision at the proofing stage.

With best regards,

Stefano Merzi

on behalf of the authors.

Reviewer 2 Report

Comments and Suggestions for Authors

1. Do not use point to replace X, for example, 8 x 108 p/cm2.

2. Please check the international units.

3. After irradiation, how about the changes of composition and microstructure?

4. It is difficult to distinguish the curves in Figs. 4, 5, 6. It is better for author to revise them. 

Comments on the Quality of English Language

The language needs to be polished. 

Author Response

Dear Reviewer,

Thank you very much for your work and for the opportunity to improve and refine our manuscript.
We addressed and accepted all the comment. Below we outline our response to each comment.

Comment 1: Do not use point to replace X, for example, 8 x 108 p/cm2.

Response 1: Points are replaced with multiplication symbols.

Comment 2: Please check the international units.

Response 2: We converted degree Celsius to Kelvin. Other units (such as Gy/year) are commonly used in the field and are kept as they are. Please advise if there are more units to convert.

Comment 3: After irradiation, how about the changes of composition and microstructure?

Response 3: No change in composition and microstructure happen at these radiation levels and it was never observed in the paper used as reference on effects of irradiation of SiPMs, either for space application or for high energy physics applications. We believe that the fluences used in our investigation are not high enough to see such effects.

Comment 4: It is difficult to distinguish the curves in Figs. 4, 5, 6. It is better for author to revise them.

Response 4: Unfortunately, most points coincide in the curves and the logarithmic scale is necessary to compare the effects at different radiation levels. We tried to differentiate the curves by setting different colors and different point styles. We also removed most of the lines for clarity reasons, but we left the most relevant devices.

We increased the size of the plots to improve readability.

We improved the English of the text as suggested.
Moreover, we think that as this paper is submitted as an open access in a special issue, it will undergo a final English revision at the proofing stage.

With best regards,

Stefano Merzi

on behalf of the authors.

Reviewer 3 Report

Comments and Suggestions for Authors

Good work

Author Response

Dear Reviewer,

Thank you very much for your work. We uploaded a revised version of the manuscript with some additional discussion and descriptions, and some polishing of the language and of the plots.

With best regards,

Stefano Merzi

on behalf of the authors.

Reviewer 4 Report

Comments and Suggestions for Authors

The Article is devoted to the study of changes in the parameters of Silicon Photomultipliers caused by their irradiation with protons and X-rays of varying intensities. Since Silicon Photomultipliers are currently finding important applications in various fields, including near-Earth space missions, medical imaging (etc.) the research topic seems relevant and important. The work studied various types of Silicon Photomultipliers (32 structures) and showed the effect of irradiation on various parameters, such as  breakdown voltage and reverse current, dark count rate, damage factor, breakdown voltage, reverse current. The methods chosen to conduct the study are consistent with the objectives of the study. The conclusions are confirmed by the given experimental data. This defines the strength of the work. The weak point is the presence of stylistic errors, which in some cases cause difficulty in understanding, the presence of a large number of unexplained abbreviations in the text, as well as the poor placement of Figures in the text. The comments are presented in the attached pdf file. The bibliography contains references to works of recent years and is not overloaded with self-citation.

The work present new results, provide useful recommendations. The Article is of interest to the journal and can be published with minor improvements.

Comments on the Quality of English Language

Minor editing of English language required

Author Response

Dear Reviewer,

Thank you very much for your work and for the opportunity to improve and refine our manuscript.
All comments are accepted and addressed in the revised manuscript, with change tracking.

The main changes are the following:

  • We added explanation of the abbreviations used in the manuscript
  • We modified the second paragraph of section 1 to improve clarity and reduce complexity of the sentences
  • We added an explanation in Figure 3 regarding the designation of PCB A, B, and C
  • We numbered formulas
  • We modified part of the conclusions to improve clarity and readability

With best regards,

Stefano Merzi

on behalf of the authors.